# Novel Insights into the Biochemical Mechanism of CK1ε and its Functional Interplay with DDX3X

**DOI:** 10.3390/ijms21176449

**Published:** 2020-09-03

**Authors:** Bartolo Bono, Giulia Franco, Valentina Riva, Anna Garbelli, Giovanni Maga

**Affiliations:** Institute of Molecular Genetics IGM-CNR “Luigi Luca Cavalli-Sforza”, via Abbiategrasso 207, 27100 Pavia, Italy; bartolo.bono@igm.cnr.it (B.B.); giulia.franco01@universitadipavia.it (G.F.); valentina.riva01@universitadipavia.it (V.R.); anna.garbelli@igm.cnr.it (A.G.)

**Keywords:** protein kinase, CK1ε, DDX3X, ATP, kinetic analysis

## Abstract

Casein Kinase 1 epsilon (CK1ε) is a member of the serine (Ser)/threonine (Thr) CK1 family, known to have crucial roles in several biological scenarios and, ever more frequently, in pathological contexts, such as cancer. Recently, the human DEAD-box RNA helicase 3 X-linked (DDX3X), involved in cancer proliferation and viral infections, has been identified as one of CK1ε substrates and its positive regulator in the Wnt/β-catenin network. However, the way by which these two proteins influence each other has not been fully clarified. In order to further investigate their interplay, we defined the kinetic parameters of CK1ε towards its substrates: ATP, casein, Dvl2 and DDX3X. CK1ε affinity for ATP depends on the nature of the substrate: increasing of casein concentrations led to an increase of Km^ATP^, while increasing DDX3X reduced it. In literature, DDX3X is described to act as an allosteric activator of CK1ε. However, when we performed kinase reactions combining DDX3X and casein, we did not find a positive effect of DDX3X on casein phosphorylation by CK1ε, while both substrates were phosphorylated in a competitive manner. Moreover, CK1ε positively stimulates DDX3X ATPase activity. Our data provide a more detailed kinetic characterization on the functional interplay of these two proteins.

## 1. Introduction

The casein kinase 1 (CK1) family is a group of multifunctional Ser/Thr-specific kinases ubiquitously expressed in eukaryotes, from yeasts to human, and involved in the regulation of several biological processes, such as cell proliferation and differentiation, chromosome segregation, DNA repair, cell survival and apoptosis, and the circadian rhythms [1,2,3,4,5]. Historically, CK1 enzymes represent one of the first group of kinases to be identified and take the name from their catalytic activity on casein proteins [6]. So far, seven *CSNK1* genes encoding distinct isoforms (α, β, γ1, γ2, γ3, δ, ε) and various related splice variants for CK1α, γ3, δ and ε have been identified and characterized in vertebrates.

All CK1 isoforms are structurally organized in a single monomer which is composed by a highly conserved central kinase domain; whereas, they differ at the level of their short amino (N)-terminus and, especially, at the carboxyl (C)-terminus that is involved in substrate recognition, regulation of catalytic activity and subcellular localization [5,7,8]. Adding to the complexity of the roles fulfilled by this class of enzymes is their wide spectrum of substrate specificity. In fact, many studies have shown that CK1 members phosphorylate about 50% of targets within the human phosphoproteome [9,10,11]. For this reason, it is not surprising that various isoforms are involved in regulating and controlling many cellular processes, as well as that the deregulation of the activity of these enzymes was demonstrated to be associated with many disorders, including tumors and neurodegenerative diseases [8,12]. Within the CK1 family, the CK1ε isoform is a key regulator of several biological processes, such as the circadian clock system, vesicles trafficking and DNA replication and repair [2,13,14]. Deregulation of CK1ε has also been reported in pathological conditions, such as several types of cancers and neurodegenerative diseases [12,15,16,17,18].

Members of CK1 family, including CK1ε, have been frequently reported to be involved in the Wnt/β-catenin network, which is one of the crucial signaling pathway associated with the initiation and progression of many tumors [19]. A large body of evidence shows that CK1ε is an activator of Wnt signaling cascade [20,21,22]. Recent studies have reported that this role seems to be played in cooperation with the human DDX3X protein, a DEAD-box ATPase/RNA helicase acting in several molecular pathways associated with normal biological processes, such as RNA metabolism, transcription, translation, cell proliferation, innate immunity, as well as pathological scenarios, as cancer, neurodegenerative diseases and viral infections [23,24,25]. Particularly, DDX3X has been reported to act as substrate and a positive regulator of CK1ε, enhancing its kinase activity on Dvl (Dishevelled) proteins which in turn induce β-catenin stabilization and nuclear translocation [26].

To date, specific therapeutic treatments to impair Wnt/β-catenin-associated tumorigenesis of several cancers are very few. To this regard, CK1 enzymes may represent molecular targets to contrast the aberrant activation of this pathway. Among CK1 members, during the last years CK1ε alterations have increasingly been considered relevant for tumor growth, becoming a promising anticancer target [8,16,27,28]. Since both CK1ε and DDX3X play several roles, sometimes controversial, in numerous pathological contexts, a precise molecular characterization of their interplay is important to develop novel therapeutic approaches.

In the present study we aimed to investigate the kinetic mechanism in vitro of CK1ε towards its novel substrate DDX3X, in comparison to its known substrates casein and Dvl2. We demonstrated that CK1ε follows a similar kinetics of random binding towards casein and DDX3X substrates, showing however a greater catalytic efficiency when saturated with ATP, while it apparently preferentially binds Dvl2 first, followed by ATP. We also confirmed that CK1ε specifically phosphorylates DDX3X at C-terminus level, while, in contrast with previously reported data, DDX3X does not appear to enhance the activity of the kinase in vitro, suggesting that additional cellular factors might modulate the mechanism of interaction between the two enzymes.

## 2. Results

### 2.1. Determination of the Optimal Initial Conditions of the Reaction Catalyzed by CK1ε

CK1ε is a Ser/Thr kinase involved in the phosphorylation of many substrates. However, the kinetic characterization of CK1ε against its conventional substrate casein is still not fully investigated. In order to perform this analysis, we took advantage of a human recombinant active full length CK1ε (SignalChem, Richmond, BC, Canada) and a mixture of dephosphorylated native casein proteins (α, β and κ isoforms) as substrates, and we assessed the enzyme activity measuring the incorporation of the γ-phosphate derived from radioactive ATP molecules. All Ser/Thr kinase reactions were performed according to manufacturer’s instructions. Initially, preliminary settings were conducted with the aim to define the optimal enzyme amount and the time of reaction. In these preliminary reactions, the initial substrate and ATP concentrations were maintained constant at 3.12 μM and 30 μM, respectively (Figure 1A,B).

The variation of the enzymatic activity (reaction velocity, v) is represented in the graphs. According to the results, we have selected an enzyme concentration of 2.5 ng/μL, (Figure 1A), and 15 min as incubation time since the increase of enzyme activity in this time interval was still linear (Figure 1B). Next, we sought to determine the kinetic parameters for the reaction.

### 2.2. Kinetic Analysis of CK1ε Acting on ATP and Casein

All kinase enzymes are characterized by the ability to bind two substrates: a molecule of ATP and a target sequence containing the specific tyrosine, serine, or threonine residues to be phosphorylated. An enzyme acting on two substrates, in principle can reach its catalytically competent ternary complex through two alternative pathways, each implying association of one substrate first, followed by the second one. If the two pathways are kinetically equivalent, that is the binding of one substrate does not influence the association of the other, then the enzyme follows a random sequential mechanism of substrate binding. Alternatively, if binding of one substrate is favored over the other, which in turn preferentially associates with the binary complex thus formed, then the enzyme follows a preferential order of binding. These two mechanisms can be discriminated by studying the effects of one substrate on the affinity of the enzyme for the second one. Any variation of the maximal velocity rate or V_max_, or the Michaelis–Menten constant or K_m_, also called affinity constant, which defines the substrate concentration at which the reaction rate is half of V_max_, for one substrate as a function of the other, might indicate a preferential order of substrate addition. Thus, we investigated whether the enzyme binds casein and ATP in a preferred manner. To this regard, the V_max_ and K_m_ values for the transfer reaction were measured by titrating ATP in parallel reactions, each in the presence of different fixed concentrations of casein (Figure 2A), and vice versa (Figure 2B). The results were then analyzed according to Michaelis–Menten equation.

The apparent affinity (K_m_) for both ATP and casein was in the low micromolar range (Table 1 and Table 2). We then derived the k_cat_ values from the starting enzyme concentration and the V_max_ (see Materials and Methods). The k_cat_ represents the limiting rate of enzyme-catalyzed reactions and is expressed as the quantity of substrate turned into product, per catalytic site per minute (min^−1^). Increasing concentrations of casein led to a moderate decrease in the affinity of CK1ε towards ATP, since the K_m_^ATP^ values increased at saturating levels of casein (Figure 2A and Table 1). On the other hand, when the reciprocal experiment was carried out varying casein at fixed ATP doses, the K_m_^casein^ values (Figure 2B and Table 2) were not influenced by increasing concentration of ATP, suggesting that ATP binding did not impact the affinity of CK1ε for casein. As shown in Table 1, increasing casein concentrations accelerated the turnover of ATP almost 10-fold, as demonstrated by the increase in k_cat_. A similar, but more modest effect was observed for casein phosphorylation (Table 2). We have then determined the catalytic efficiencies (k_cat_/K_m_) of CK1ε in both experimental conditions. We found that catalytic efficiency increased at increasing substrate concentrations, for both substrates (Table 1 and Table 2). At saturating ATP concentrations, the efficiency of the association of CK1ε to the casein substrate (k_cat_/K_m_^casein^) was almost 3.5-fold higher than towards the ATP substrate (k_cat_/K_m_^ATP^) at saturating casein (Table 1 and Table 2). Since the influence of each substrate on the affinity of the other was modest, all together these results do not indicate a preferential order of substrate binding. However, they suggest that saturation of the enzyme with ATP increases the rate of association of casein.

### 2.3. Interplay of CK1ε and the Human DDX3X Recombinant Protein in the Phosphorylation Reaction

It has been reported that DDX3X interacts with CK1ε acting as a substrate for the kinase [26]. To provide a more detailed kinetic study on the interplay between these proteins, we exploited a recombinant human DDX3X purified in our laboratory, and we used it as substrate of CK1ε in kinase assays. To investigate the kinetic mechanism of CK1ε activity towards the substrate DDX3X, we have firstly determined the variation of the velocity reaction rate as a function of varying DDX3X or ATP concentrations, keeping the reciprocal substrate fixed (Figure 3A,B). We used fixed DDX3X and ATP concentrations as 0.48 μM and 30 μM, respectively. From these experiments we determined the V_max_ and K_m_ values for both reaction conditions (Table 3). Our data indicated that CK1ε has high affinity for DDX3X (0.2 µM). Since in our laboratory the mean concentrations of DDX3X in various cell lines have been determined in the range of 100–700 nM [29], the calculated affinity makes DDX3X a substrate of CK1ε at its physiological concentrations. Thus the reciprocal influence of DDX3X and ATP was investigated in more detail.

To this aim, we have performed a kinetic study of CK1ε vs DDX3X and ATP substrates, comparing the obtained values of V_max_, K_m_, and k_cat_/K_m_ parameters at fixed DDX3X concentrations, by varying the ATP amount, and vice versa. At the highest DDX3X concentration of 0.96 µM the affinity for ATP slightly increased (K_m_: 16.2 µM) (Figure 4A and Table 4). On the other hand, increasing the ATP fixed concentrations did not influence the K_m_ of CK1ε for DDX3X (Figure 4B and Table 5). This indicates again a random sequential order of substrate binding. We have then calculated the k_cat_/K_m_ in both conditions. Both substrates increased the catalytic efficiency for the reciprocal one: DDX3X increased the efficiency for ATP hydrolysis by 4-fold (Table 4), while ATP increased the k_cat_/K_m_ for DDX3X phosphorylation by approximately 3.5-fold (Table 5). However, while the k_cat_/K_m_ for DDX3X at saturating ATP concentrations (Table 5) was comparable to the one observed with casein (Table 2), the CK1ε catalytic efficiency towards ATP with DDX3X as a substrate (Table 4) was 8-fold lower than those observed at saturating casein (Table 1) (3.1 vs 0.37 min^−1^ μM^−1^). Together these data suggest that casein and DDX3X are recognized as substrates with similar efficiencies, but the turnover rate of ATP hydrolysis was lower with DDX3X than with casein.

### 2.4. Kinetic Analysis of CK1ε Acting on ATP and the Dvl2 Protein

Dvl2, known as segment polarity protein dishevelled 2, is a member of the Dvl scaffold proteins and plays a key role in the Wnt/β-catenin signaling pathway [30]. This protein can also be phosphorylated by CK1ε, resulting in an enhancement of the activation of the pathway, as reported in colorectal cancer [22]. Given that Dvl2 is also substrate of CK1ε, we performed the same kinetic studies previously conducted for casein and DDX3X. To investigate it, we exploited a human recombinant Dvl2 (Abnova, Taipei City, Taiwan).

We then determined the kinetic parameters of CK1ε reaction as a function of varying ATP at fixed concentrations of Dvl2, or vice versa (Figure 5A,B). Varying the ATP amount at different fixed concentrations of Dvl2, the affinity for ATP increased (decrease in K_m_ values) as Dvl2 concentration increased, and the k_cat_ value increased (Figure 5A, Table 6). Thus, the efficiency of the enzyme in hydrolyzing ATP (k_cat_/K_m_) increased from 1.0 to 7.6 min^−1^ μΜ^−1^ by saturation of the enzyme with Dvl2 (Table 6). In the reciprocal reactions, the K_m_^Dvl2^ values were generally not influenced by the increasing concentration of ATP (Figure 5B, Table 7), suggesting that ATP binding did not impact the affinity of CK1ε for Dvl2. Thus, it appears that contrary to what we observed for casein and DDX3X, binding of ATP to a binary Ck1ε-Dvl2 complex would be the preferred kinetic pathway. Interestingly, the CK1ε catalytic efficiency was notably higher when the enzyme was saturated with Dvl2 rather than casein (Table 7 and Table 2), thus CK1ε seems to better catalyze the Dvl2 phosphorylation with respect to casein, suggesting that Dvl2 might a more specific substrate for Ck1ε, according to its role in the Wnt/β-catenin signaling pathway.

### 2.5. CK1ε Phosphorylates Casein and DDX3X in a Competitive Way

Recently, DDX3X was shown to function as a regulatory subunit of CK1ε in the Wnt/β catenin signaling pathway. The interaction with DDX3X enhanced the CK1ε activity in vitro, thus suggesting a role of DDX3X as a positive allosteric modulator [26]. In order to study more in details if the CK1ε activity was influenced by DDX3X, we compared the kinase activity towards casein in presence or absence of recombinant DDX3X. We performed kinase reactions in presence of γ^33^P-ATP and we then stopped and loaded them on SDS-PAGE and finally transferred on nitrocellulose membrane. Phosphorylated proteins were visualized by autoradiography (Figure 6A, upper panel). Loading control was performed for CK1ε and DDX3X by western blotting (Figure 6A, lower panels). To evaluate the effect of the presence of DDX3X on casein phosphorylation by CK1ε, we used three different concentrations of casein (0.13, 0.06 and 0.03 μM) and a fixed DDX3X concentration (1.5 μM), and we compared it to casein phosphorylation alone, used as control. In Figure 6A, autoradiography shows the detection of a series of bands corresponding to the phosphorylated casein (around 25–30 kDa) (P-casein) and recombinant DDX3X (around 95–100 kDa) (P-DDX3X). Moreover, it is possible to notice the presence of a band of about 72 kDa, corresponding to autophosphorylated CK1ε (AutoP-CK1ε) (Figure 6A, upper panel) as confirmed by immunoblot (Figure 6A, lower panel). Quantitative analysis of the different phosphorylated products relative to P-casein (Figure 6A, lanes 1-6), showed a decrease of the intensity of the P-casein signals in the presence of DDX3X (Figure 6A, lanes 4–6) compared to controls (Figure 6A, lanes 1–3). This observation was confirmed by the quantification of P-casein relative to the total phosphorylated products (Total P) (Figure 6B, top panel). Analyzing the ratios between P-DDX3X and Total P, we also observed a reduced P-DDX3X amount in the presence of casein (Figure 6B, middle panel) compared to control with DDX3X alone (Figure 6B, middle panel). We have also quantified the signal of AutoP-CK1ε which was decreased in all reactions (Figure 6B, bottom panel) compared to control in the absence of substrates (Figure 6B, bottom panel and 6A, lane 8). These findings confirm that CK1ε interacts with and phosphorylates DDX3X, but the presence of DDX3X under our in vitro conditions did not enhance CK1ε activity on casein. The data, indeed, are consistent with the two substrates competing with each other for the kinase active site. The reduction observed in the AutoP-CK1ε in the presence of the different substrates, is likely the consequence of the kinetic partition of ATP hydrolysis among the different reaction pathways leading to AutoP-CK1ε, P-casein, and P-DDX3X, respectively.

It has been demonstrated that CK1ε phosphorylates DDX3X on specific Ser/Thr residues at C-terminal [31]. To investigate whether also in our kinase assays CK1ε phosphorylates DDX3X at the C-terminal side, we performed a kinase reaction using a C-terminal truncated (Δ428-662 aa) DDX3X (N-DDX3X, ~50kDa) and we analyzed phosphorylation by autoradiography (Figure 6C, upper panel). We confirmed the presence of N-DDX3X by western blotting (Figure 7C, lower panel). We did not detect a signal around 50 kDa, as eventually expected in case of N-DDX3X phosphorylation (Figure 6C, lane 7), confirming that CK1ε phosphorylates DDX3X at its C-terminal alone. In addition, in order to investigate a possible positive effect of the presence of N-DDX3X on CK1ε activity towards casein, we combined casein at different doses (0.13, 0.06 and 0.03 μM) with a fixed amount of N-DDX3X (1.5 μM) in the CK1ε reactions (Figure 6C, lanes 4–6), and we compared it to controls with casein alone (Figure 6C, lanes 1–3). Results showed only the bands corresponding to the P-casein (around 25–30 kDa) or AutoP-CK1ε (around 72 kDa) (Figure 6C). The intensity of P-casein signals increased in a concentration-dependent manner (Figure 6C, lanes 1–3 and 4–6). We then quantified the P-casein signals respect to Total P in each reaction, but we did not find differences between the phosphorylation ratios of the combined reactions (Figure 6D,C lanes 4–6) and controls (Figure 6D,C lanes 1–3), indicating that N-DDX3 does not appear to positively affect CK1ε activity. Moreover, we also performed a quantitative analysis of the levels of AutoP-CK1ε and, as shown in Figure 6D, N-DDX3X did not affect the AutoP-CK1ε levels, again confirming that it was not a substrate for the enzyme, while AutoP-CK1ε levels were markedly reduced in all reactions containing casein, compared to controls (Figure 6D,C lane 8), suggesting again that substrates compete with the enzyme for the binding to the active site. Similar experiments were performed combining CK1ε, DDX3X, and Dvl2, but we did not detect any stimulation of CK1ε activity towards Dvl2 by DDX3X (Appendix A).

### 2.6. CK1ε Positively Modulates the DDX3X ATPase Activity

Next, we focused our attention on the possible role of the kinetic interplay between CK1ε and DDX3X, on the ATPase activity of DDX3X. It has been suggested that CK1ε reduced the dsRNA-dependent DDX3X ATPase activity by phosphorylating DDX3X [31]. Here, we investigated whether the basal ATPase activity of the human recombinant DDX3X, could be affected by CK1ε-dependent phosphorylation. We firstly assessed the basal ATPase activity of DDX3X (for the sake of simplicity here called wild type, WT) in comparison with that of a specific DDX3X mutant (DDX3X DADA) created in our laboratory, which was specifically modified in two peculiar residues, Asp346Ala/Asp349Ala (DA/DA), of the DEAD box important for ATP binding and hydrolysis [32]. The ATPase activity was measured as the luminescence in relative light units (RLU) (see Materials and Methods). Since the values of RLU are proportional to the amount of ADP produced after the ATP hydrolysis, the results of the assay revealed that the WT enzyme is endowed with a higher ATPase activity than the DADA mutant, as expected (Figure 7A).

To investigate the ATPase activity of phosphorylated DDX3X (P-DDX3X), DDX3X was firstly incubated with CK1ε to induce its phosphorylation and then assessed for the ATPase activity. In order to eliminate the potentially interfering ATPase activity of CK1ε, due to its kinase activity, we used a CK1ε-specific inhibitor, PF4800567 [33,34,35]. Initially, we performed a preliminary screening of the PF4800567compound with the aim to determine the ID_50_ dose. The compound was assayed in the phosphate incorporation reactions on casein or DDX3X (Figure 7B). Analyzing the reaction velocity of CK1ε towards casein as function of several PF4800567 doses, we determined an ID_50_ value of 162.2 nM, while the value was around 1.1 μM towards DDX3X phospho-incorporation (Figure 7B). On the basis of these results, in order to have a negative baseline of ATPase activity of CK1ε during the measurement of the P-DDX3X ATPase assay, we decided to use a dose of inhibitor 13-fold greater than the observed ID_50_. As shown in Figure 7C, in a first experimental set-up, phosphorylated P-DDX3X had significantly increased ATPase activity compared to the unphosphorylated controls (DDX3X, DDX3X B) (*p*-value = 0.02 and 0.04). However, in order to exclude a contribution to ATP hydrolysis possibly derived from CK1ε activity during the ATPase reaction step, we induced DDX3X phosphorylation with the kinase reaction and we then added the CK1ε inhibitor PF4800567 (P-DDX3X A) before running the ATPase assay. In these conditions, we could observe a significant, even higher, enhancement of the phosphorylated DDX3X ATPase activity (P-DDX3X A) compared to the unphosphorylated forms (DDX3X, DDX3X B) (*p*-value = 0.01 and 0.03). In addition, further controls without DDX3X allowed us to exclude a possible background deriving from autophosphorylation of CK1ε. In fact, no relevant ATPase activity was detected in the presence of CK1ε alone (Figure 7C). These results suggested that phosphorylation of DDX3X by CK1ε enhances the basal ATPase activity of DDX3X.

## 3. Discussion

Among the members of the Serine/Threonine CK1 family, CK1ε is able to transfer γ-phosphate groups from ATP to specific Ser and Thr residues of various substrates [5] and represents an important regulator of key molecular pathways whose dysregulation has been frequently reported in several type of cancers, as well as neurodegenerative diseases [36,37,38]. Recent findings have reported that this kinase plays an important role in the induction of the well-known Wnt/β-catenin signaling cascade and, particularly, this role seems to be supported by its functional interaction with the ATP-dependent RNA helicase DDX3X, a multifunctional protein involved in several normal and pathological contexts, including viral infection and neoplastic transformation. Specifically, DDX3X was demonstrated to be a regulator of the Wnt network by the scaffold protein Dvl2 in mammalian cells and during *Xenopus* and *Caenorhabditis elegans* development [26]. Moreover, another study has revealed that the interplay between CK1ε and DDX3X has also a role in the insurgence of neurodegenerative diseases, as in amyotrophic lateral sclerosis (ALS) [17].

In this work we aimed at characterizing from a kinetic point of view their reciprocal interplay in vitro. The kinetic analysis performed in the first part of this project indicated that CK1ε does not appear to follow an obligatory sequential binding of substrates. Thus, our observations seem consistent with studies that had already been on other Ser/Thr kinases, such as PKA and AKTs, in spite of several differences in their molecular structure compared to CK1 family members [39,40]. However, binding of ATP increases the apparent limiting association rate (k_cat_/K_m_) of casein. A similar kinetic behavior was observed for DDX3X, which has been identified among the CK1ε substrates that are involved in Wnt signaling [26,31]. Considering the high affinity for ATP, as compared to its physiological concentrations, it is likely that at equilibrium in vivo the binary enzyme-ATP complex would form faster, which will be than ready to associate with the second substrate of the kinase. We also analyzed another physiological substrate of CK1ε, the protein Dvl2. In this latter case, the kinetic data suggested that binding of ATP to the binary CK1ε-Dvl2 complex would be the most favored pathway. It is noteworthy that the apparent limiting association rate (k_cat_/K_m_) to Dvl2 was 10-fold higher than for casein. These results suggest that Dvl2 is a very specific substrate for CK1ε, ensuring always maximal efficiency of phosphorylation under physiological conditions. This is consistent with the essential role of Ck1ε phosphorylation of Dvl2 in the Wnt signaling pathway.

The kinetic constant values were also different comparing DDX3X to casein and Dvl2, thus suggesting different catalytic interactions. The main differences were associated with the affinity of CK1ε for DDX3X, which was intermediate between casein and Dvl2 but still within the physiological range of DDX3X intracellular concentrations, and especially for ATP hydrolysis at DDX3X saturating amount, which was much lower than the one observed with casein or Dvl2. The slow rate of ATP turnover by CK1ε upon DDX3X binding, might be part of a fine-tuned regulatory mechanism, whereby additional cellular factors and/or post-translational modifications might increase the rate DDX3X phosphorylation by CK1ε.

In parallel, we have directly visualized the CK1ε kinase activity through the autoradiographic detection of incorporated radiolabeled phosphate. It has been reported that amino acids Ser323, Thr325, Thr334, Thr337, Ser368, Ser405, Thr407, and Ser408 in the carboxyl-terminal tail of CK1ε are sites of auto-phosphorylation, that may reduce its kinase activity [10]. We firstly confirmed that the enzyme was able to auto-phosphorylate itself and that in our assay conditions the enzyme remains in the active status. Interestingly, we observed that DDX3X phosphorylation was decreased by increasing casein concentrations and vice versa. In addition, both casein and DDX3X reduced CK1ε autophosphorylation. Thus, both substrates compete for CK1ε binding and reflect a partition of ATP hydrolysis among the three mutually exclusive kinetic pathways (CK1ε autophosphorylation, casein phosphorylation, DDX3X phosphorylation). DDX3X has been reported to directly interact and to promote the activity of some kinases, such as TBK1, IKKε, and CK1ε [26,41,42]. However, here we did not observe an increase of the kinase activity upon addition of DDX3X, indicating that in our in vitro conditions DDX3X acts as a substrate, not as an activator. This may suggest that the function(s) of DDX3X as an activator of CK1ε in vivo could involve additional co-factors, which we are currently aiming at identifying.

Dolde and colleagues recently showed that CK1ε phosphorylates DDX3X in several sites at the C-terminus [31]. Thus, in order to investigate whether our in vitro system recapitulates the known phosphorylation pattern of recombinant DDX3X, we performed a kinase reaction using a C-terminal truncated DDX3X form (N-DDX3X, ~50 kDa). We did not detect N-DDX3X phosphorylation, confirming that in our case, too, DDX3X is phosphorylated at its C-terminus. Furthermore, contrary to what has been observed with casein, CK1ε autophosphorylation was not reduced by the presence of N-DDX3, which might suggest that DDX3X requires an intact C-terminus for physical interaction with CK1ε. This is currently under investigation.

Beyond the kinetic characterization of CK1ε towards DDX3X, we have also investigated whether the CK1ε kinase activity could affect the ATPase function of DDX3X. It is well-known that DDX3X is a RNA helicase that requires ATP to reach the proper conformation and unwind double stranded nucleic acids [31,32,43]. Dolde and colleagues had demonstrated that CK1ε impairs the dsRNA-dependent DDX3X ATPase by phosphorylation [31]. On the other hand, in our in vitro system, phosphorylation of DDX3X by CK1ε enhanced its ATPase activity compared to the non-phosphorylated one. One difference among our assay and the one used by Dolde and colleagues was the choice of the nucleic acid used to stimulate DDX3X ATPase. While in the previous study it has been used a dsRNA oligonucleotide, here we used a ssDNA. dsRNA is a substrate for the helicase activity. Crystal structure of the active complex of DDX3X with an RNA substrate has shown that DDX3X binds dsRNA as a dimer [44]. Multiple cycles of ATP binding and hydrolysis by the two monomers are then required during the unwinding reaction. On the other hand, ssDNA, while still able to stimulate ATP hydrolysis [32], is not a substrate for the helicase reaction. According to the crystal structure [44], a ss nucleic acid lattice can be bound by DDX3X only as a monomer, thus the kinetic of the ATPase reaction is not influenced by enzyme stoichiometry and/or concomitant unwinding of the substrate. It is possible that phosphorylation of DDX3X affects its translocation/unwinding along the dsRNA lattice and/or its stoichiometry of binding, rather than the ATPase activity per se. Further analysis of a broad range of nucleic acids cofactors might better clarify these points. In the future, we will aim at understanding if DDX3X helicase activity could be influenced by phosphorylation and which could be the consequences of such post-translational modification.

## 4. Materials and Methods

### 4.1. Chemicals

Labeled [γ-^33^P]ATP (3000 Ci/mmol) was purchased from Hartmann Analytic GmbH (Braunschweig, Germany). Unlabeled ATP was from Sigma-Aldrich (St. Louis, MO, USA). The PF4800567 compound was developed and provided by Calbiochem (San Diego, CA, USA). All other general reagents were from Sigma-Aldrich (St. Louis, MO, USA) and Merck KGaA (Darmstadt, Germany).

### 4.2. Enzymes and Proteins

Unless specified otherwise, full length recombinant human active CK1 epsilon (CK1ε) and dephosphorylated native casein mixture (Casein) were purchased from SignalChem (Richmond, BC, Canada), while the full length recombinant human disheveled 2 (Dvl2) were purchased in the highest available quality from Abnova (Taipei City, Taiwan). All proteins were used and conserved according to the manufacturer’s instructions. The human DDX3X recombinant proteins were purified in our lab. The N-DDX3X and DDX3X DADA mutant were obtained as described [32,45]. The recombinant DDX3X wild-type (WT) was expressed in the *Escherichia coli* K12 shuffle T7 strain. Using the pET-30a(+) plasmid vector (Merck KGaA, Darmstadt, Germany), the recombinant protein has two 6xHis affinity tags, upstream and downstream of the MCS sequence respectively. Transformed bacteria was then selected with kanamycin (50 μg/mL) (AppliChem GmbH, Darmstadt, Germany) and then grown in agitation at 37 °C overnight (O/N) in Luria Both (LB) medium. The obtained pre-inoculum was transferred into growth medium complemented with 50 μg/mL of kanamycin and 0.01% anti-foam (Sigma-Aldrich, St. Louis, MO, USA) and grown up at 37 °C until to optical density (OD_600_) around 0.6–0.8. DDX3X expression was then induced by 0.5 mM IPTG (Sigma-Aldrich, St. Louis, MO, USA) in agitation at 15 °C O/N. Bacterial pellet was then collected and resuspended on ice in lysis buffer (10 mM Tris HCl pH 8, 100 mM Na_3_PO_4_ pH 8, 10% NP-40) complemented with 1 mg/mL lysozyme, 1 mM PMSF and 1X protease inhibitor cocktail (Sigma-Aldrich, St. Louis, MO, USA) for 30 min and finally sonicated for 5 min. The lysate was then ultracentrifuged at 38,000× *g* in a Beckman centrifuge for 1 h at 4 °C and the supernatant was then passed through an affinity FPLC-NiNTA-column (GE Heatlthcare Life Science, Marlborough, MA, USA), which was previously equilibrated with Buffer A (50 mM Tris HCl pH 8, 250 mM NaCl, 25 mM imidazole, 20% glycerol). Proteins bound to the column were eluted with a linear gradient (25–250 mM imidazole) in Buffer A, exploiting a Fast Protein Liquid Chromatography (FPLC) System (Bio-Rad Laboratories, Hercules, CA, USA). The collected fractions were visualized and checked for the presence of pure proteins by Coomassie Blue staining and using an anti-DDX3X antibody (A300–475A, Bethyl Laboratories, Montgomery, TX, USA) by Western blotting. Fractions containing purest DDX3X proteins were pooled and dialyzed (25 mM Tris HCl pH 8, 100 mM NaCl, 0.5 mM DTT, 20% glycerol), flash-frozen in liquid nitrogen, and stored at -80 °C.

### 4.3. Kinase Activity Assay

Kinase reactions were carried out for 15 min at 30 °C in 10 µL containing recombinant proteins (casein, DDX3X, N-DDX3X, Dvl2), used as described in the text, 0.1 µL γ^33^P-ATP (30 Ci/mmol, 10 mCi/mL), 60 μM unlabeled ATP (30 μM for preliminary set-up) in 1X Kinase Buffer I (5 mM MOPS pH 7.2, 2.5 mM β-glycerol-phosphate, 5 mM MgCl_2_, 1 mM EGTA pH 8.0, 0.4 mM EDTA pH 8, 0.05 mM DTT) and 25 ng of CK1ε active enzyme diluted in Buffer III (1X Kinase Buffer I, 50 ng/µL BSA, 5% glycerol, 0.05 mM DTT). Reactions were stopped by their transferring (9 μL) on P30 Filtermat (PerkinElmer, Waltham, MA, USA). The filter was then washed five times with 75 mM phosphoric acid for 5 min and once with acetone for 4 min, and was finally dried and transferred to a sealable plastic bag, and scintillation cocktail (4 mL) was added. Spotted reactions were read in a Trilux scintillation counter (PerkinElmer, Waltham, MA, USA). The inhibition studies with PF4800567 compound were performed using the same reaction conditions described above, and respect to casein or DDX3X substrate. The inhibitor was dissolved in 100% DMSO as indicated in manufacturing instructions, and used at 10% final concentration in control reactions. In general, this simple radioactive kinase assay was used to monitor the transfer of phosphate from [γ-^33^P]ATP to various substrates, which was detected as counts per minute (cpm) and then converted in picomoles of transferred phosphate per minute per amount of enzyme (pmol min^−1^ μg^−1^).

### 4.4. Kinetic Analysis

The determination of the kinetic parameters K_m_ and V_max_ was performed using the GraphPad Prism version 5.0 (GraphPad Software) where data were fit to the Michaelis–Menten equation (shown below) by the method of Nonlinear regression (curve fit):(1)$$v=\;V_{\max}*\;\left[S\right]/\left(K_{m}+\left[S\right]\right)$$
where [S] is the concentration of the ATP or protein substrates, V_max_ is the apparent maximal rate of the reaction and K_m_ is the Michaelis constant value for ATP or substrate.

The catalytic constant k_cat_ or turnover number is the number of substrate molecules turned into products by an enzyme molecule per unit time in enzyme saturating conditions, and was derived according to the following equation derived by GraphPad Prism version 5.0:(2)$$v=Et*\;k_{\mathrm{cat}}*\left[S\right]/\;\left(K_{m}+\left[S\right]\right)$$
where v is the apparent maximal velocity V_max_ as a function of variable ATP or substrate concentrations [S], Et is the concentration of enzyme catalytic sites, assumed equal to the total enzyme concentration [E]ₒ constrained to the constant value 0.3472 pmoles, K_m_ is the apparent affinity for ATP or substrate.

The kinetic efficiencies of substrate conversion were calculated according to the ratio:(3)$$k_{\mathrm{cat}}/K_{m}$$

The determination of the concentration of the tested compound PF4800567 required to inhibit the 50% of enzyme activity (ID_50_) was determined according to the following equation derived by GraphPad Prism version 5.0:(4)$$v=\;V_{\max}/(1+(\left[S\right]/\left[ID_{50}\right]))$$
where [ID_50_] is the concentration of PF4800567 which gives 50% inhibition.

### 4.5. Autoradiography Assay

All kinase reactions were performed as described before. Reactions were then stopped by the addition of Laemmli SDS sample dilution buffer and loaded on 10% SDS-polyacrylamide gel for protein separation by electrophoresis (SDS-PAGE). Once separated, proteins were then transferred and fixed on a nitrocellulose blotting membrane 0.45 μm (GE Healthcare Life Science, Marlborough, MA, USA). Finally, the membrane was impressed on autoradiographic sheet (Amersham plc) for about 24 h in the dark. The signal emitted from incorporated γ-^33^P (^33^P) was detected through the Typhoon-TRIO scanner device (GE Healthcare Life Science, Marlborough, MA, USA). The γ-^33^P intensity signals were quantified by the QuantityOne software (Bio-Rad Laboratories, Hercules, CA, USA). Statistical analysis was performed using Student’s *t*-test, and *p*-values lower than 0.05 or 0.01 were considered significant.

### 4.6. Western Blot Analysis

Western blotting was performed as loading control of kinase reactions on nitrocellulose blotting membrane. The antibodies used for the analysis were as follows: anti-DDX3 (C-terminal) A300-475A rabbit (Bethyl Laboratories, Montgomery, TX, USA); anti-DDX3X (N-terminal) A300-474A rabbit (Bethyl Laboratories, Montgomery, TX, USA); anti-CK1ε mouse (Abcam, Cambridge, UK); anti-rabbit or anti-mouse HRP-conjugated secondary antibodies (Jackson ImmunoResearch, Cambridge, UK). All antibodies were used at specific dilutions in TBS 1X / 5% (w/v) Skimmed Milk. Signals detection was performed using a hydrogen peroxide-luminol reaction (Cyanagen, Bologna, Italy) and the Chemidoc XRS System (Bio-Rad Laboratories, Hercules, CA, USA).

### 4.7. ATPase Activity Assay

The ATPase enzymatic activity of DDX3X and DDX3X DADA mutant was determined by the ADP-Glo^TM^ kit (Promega, Madison, WI, USA), as indicated by manufacturing instructions. The ATPase enzymatic reaction was performed at room temperature for 30 min in a final volume of 20 μL containing: 1.5 μM of the different DDX3X proteins, 0.1 mM ATP ultrapure (Promega, Madison, WI, USA), 10 μM ssDNA 32mer (Eurofins, Luxembourg), 1X Buffer (5 mM MOPS pH 7.2, 2.5 mM β-glicerol-phosphate, 10 mM MgCl_2_, 1 mM EGTA pH 8.0, 0.4 mM EDTA pH 8, 0.05 mM DTT). The ATPase enzymatic activity was analyzed using 384 wells plate (Merck KGaA, Darmstadt, Germany) and the GloMax Discover Microplate Reader (Promega, Madison, WI, USA). The ATPase activity was expressed as relative light units (RLU). Analysis of the ATPase activity of the phospho-DDX3X was performed as described in the text. Phospho-DDX3X was induced pre-incubating 1.5 μM of protein with 25 ng of CK1ε according to the previously described kinase conditions. The average of two replicates was calculated and graphically plotted using Excel. Results are presented as mean of three independent measurements ± standard deviation (SD). Statistical analysis was performed using Student’s *t*-test, and *p*-values lower than 0.05 were considered significant.

## Figures and Tables

**Figure 1 ijms-21-06449-f001:**
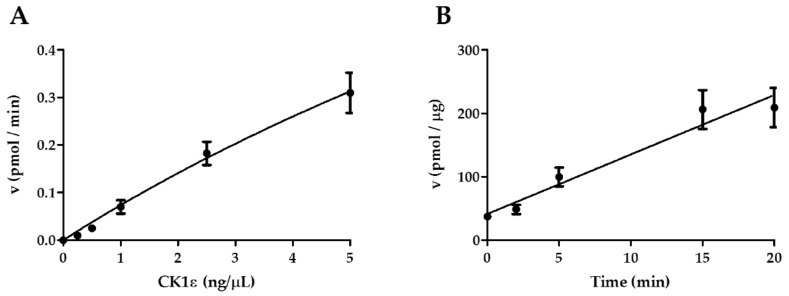
Preliminary set-up of Casein Kinase 1 epsilon (CK1ε) enzymatic activity. (**A**) CK1ε titration curve. A fixed concentration of substrate (casein) (3.12 μM) and ATP (30 μM) were incubated with serial dilutions of the enzyme (0.25–5 ng/μL) for 10 min. (**B**) Time course of CK1ε kinase activity in the presence of fixed concentration of substrate (casein) (3.12 μM) and ATP (30 μM) and of CK1ε (25 ng). Bars indicate the mean of three independent measurements ± SD (Standard Deviation).

**Figure 2 ijms-21-06449-f002:**
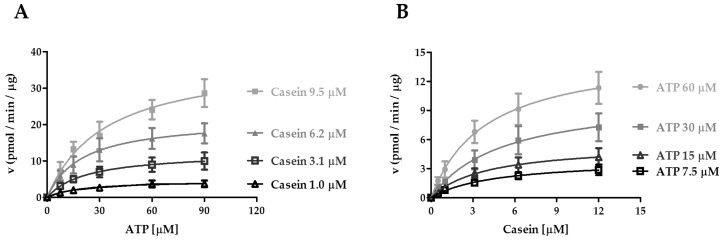
Kinetic analysis of CK1ε acting on ATP and casein. (**A**) Kinetic analysis of CK1ε towards ATP increasing concentrations at fixed casein concentrations. (**B**) Kinetic analysis of CK1ε towards casein increasing concentrations at fixed ATP concentrations. In both cases, reaction velocity rate (y axis) is measured as picomoles of incorporated phosphate per minute per microgram of enzyme (pmol/min/μg). Data are representative of three independent experiments. Bars indicate mean ± SD.

**Figure 3 ijms-21-06449-f003:**
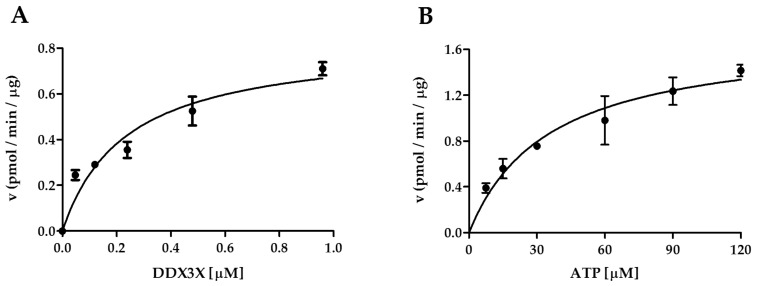
Kinetic set-up of CK1ε towars DDX3X. (**A**) Substrate titration. A fixed amount of CK1ε (25 ng) and ATP (30 μM) were incubated with increasing concentrations of DDX3X (0.06–1 μM) for 15 min. (**B**) ATP titration. A fixed amount of CK1ε (25 ng) and DDX3X (0.48 μM) were incubated with increasing concentrations of ATP (7.5–120 μM) for 15 min. Bars indicate mean ± SD.

**Figure 4 ijms-21-06449-f004:**
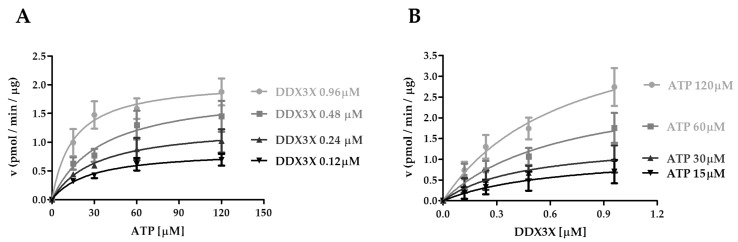
Kinetic analysis of CK1ε acting on ATP and DDX3X. (**A**) Kinetic analysis of CK1ε towards ATP increasing concentrations at fixed DDX3X concentrations. (**B**) Kinetic analysis of CK1ε towards DDX3X increasing concentrations at fixed ATP concentrations. Reaction velocity rate (y axis) is represented as picomoles of incorporated phosphate on DDX3X per minute per microgram of enzyme (pmol/min/μg). Data are representative of three independent experiments. Bars indicate mean ± SD.

**Figure 5 ijms-21-06449-f005:**
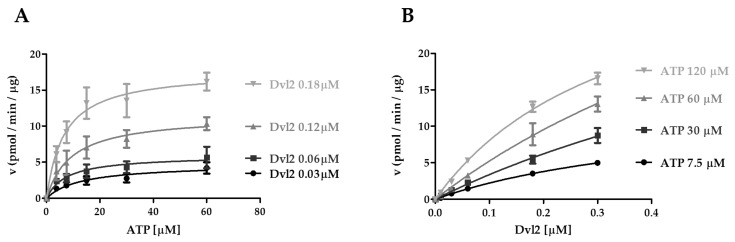
Kinetic analysis of CK1ε acting on ATP and Dvl2. (**A**) Kinetic analysis of CK1ε towards ATP increasing concentrations at fixed Dvl2 concentrations. (**B**) Kinetic analysis of CK1ε towards Dvl2 increasing concentrations at fixed ATP concentrations. Reaction velocity rate (y axis) is represented as picomoles of incorporated phosphate on Dvl2 per minute per microgram of enzyme (pmol/min/μg). Data are representative of three independent experiments. Bars indicate mean ± SD.

**Figure 6 ijms-21-06449-f006:**
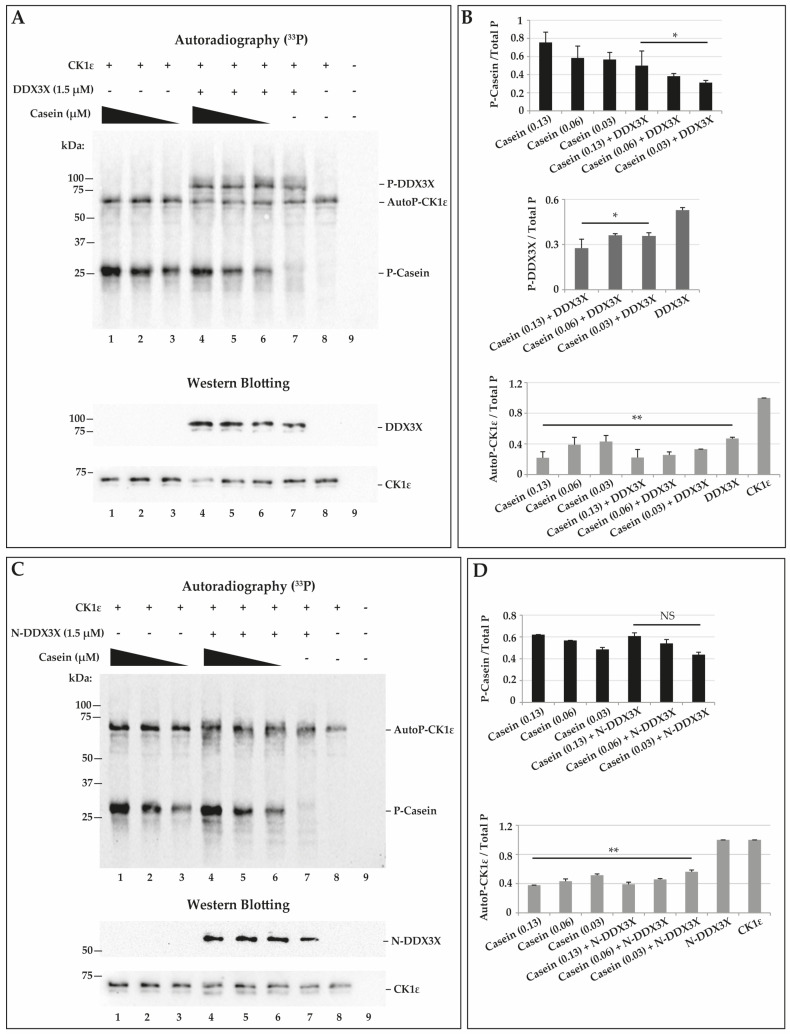
Effect of the presence of DDX3X on casein phosphorylation by CK1ε. (**A**) DDX3X, CK1ε, and casein γ-^33^P incorporation visualized by autoradiography on nitrocellulose membrane (upper panel). Western blot analysis of the same membrane (lower panels). (**B**) Quantification of autoradiographic γ-^33^P signal (panel A). Bars indicate mean ± SD. (**C**) Representation of CK1ε and casein γ-^33^P incorporation on nitrocellulose membrane visualized by autoradiography (upper panel). Western blot analysis of the same membrane (lower panel). (**D**) Quantification of autoradiographic γ-^33^P signal (panel C). Bars indicate mean ± SD. * and ** indicate *p*-value < 0.05 and < 0.01 respectively. NS: not significant.

**Figure 7 ijms-21-06449-f007:**
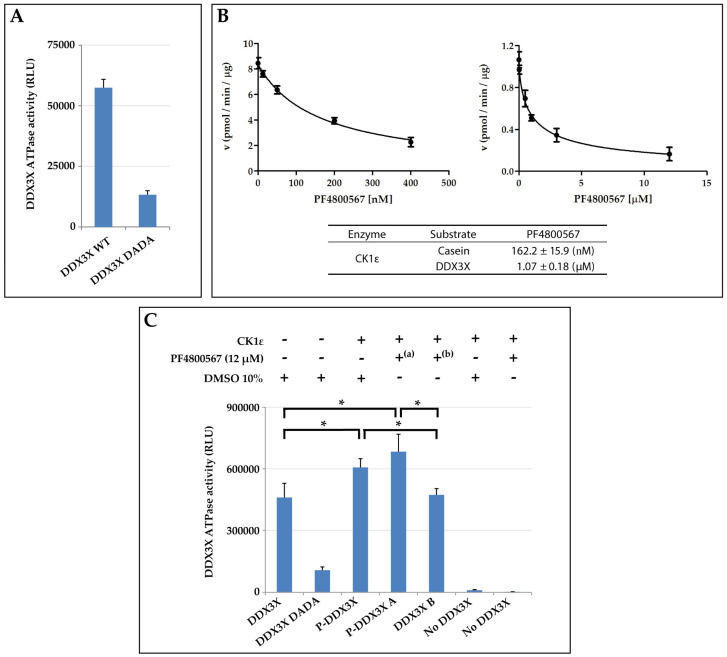
The DDX3X ATPase activity is positively influenced by the presence of CK1ε. (**A**) Definition of the basal ATPase activity of DDX3X wild-type (WT) compared to the DDX3X DADA mutant. ATPase activity is measured as relative light units (RLU). Bars indicate mean ± SD. (**B**) Inhibition curves of CK1ε activity towards casein (left) or DDX3X (right) in presence of the CK1ε-selective PF4800567 inhibitor. The table below reports the ID_50_ values (± SD). (**C**) Analysis of the ATPase activity of phosphorylated DDX3X. DDX3X ATPase activity is reported as RLU values. PF4800567 was used as inhibitor of CK1ε kinase activity. Phosphorylated DDX3X after CK1ε incubation (P-DDX3X); P-DDX3X A: phosphorylated DDX3X after CK1ε incubation, but followed by (^a^) PF4800567 addition during ATPase reaction; DDX3X B: unphosphorylated DDX3X obtained by (^b^) PF4800567 addition during CK1ε kinase reaction; DDX3X were used as positive and also unphosphorylated control of ATPase activity, while DDX3X DADA as negative control. Error bars: standard deviations. *: *p*-value < 0.05.

**Table 1 ijms-21-06449-t001:** Kinetic parameters values of CK1ε as a function of varying ATP concentrations at fixed casein amounts.

Casein [μM]	1	3.1	6.2	9.5
**V_max_^ATP^** **(pmol min^−1^** **μ** **g^−1^)**	4.7 ± 0.7 ^a^	12.4 ± 2.3 ^a^	21.6 ± 3.6 ^a^	38.2 ± 5.6 ^a^
**K_m_^ATP^** **[μM]**	21.1 ± 6.6	20.5 ± 1.2	21.1 ± 3.7	35.9 ± 7.9
**k_cat_** **(min^−1^)**	13.5 ± 2.1	35.7 ± 6.8	62.2 ± 10.3	110.1 ± 16.2
**k_cat_ / K_m_^ATP^** **(min^−1^ μM^−1^)**	0.6	1.7	2.9	3.1

^a^ All values represent the mean of three independent measurements ± SD.

**Table 2 ijms-21-06449-t002:** Kinetic parameter values of CK1ε as a function of varying casein concentrations at fixed ATP amounts.

ATP [μM]	7.5	15	30	60
**V_max_^Casein^** **(pmol min^−1^** **μ** **g^−1^)**	3.9 ± 0.4 ^a^	5.6 ± 0.7 ^a^	10.5 ± 1.4 ^a^	15.0 ± 1.4 ^a^
**K_m_^Casein^** **[μM]**	4.6 ± 0.6	3.9 ± 0.9	4.8 ± 0.8	4.0 ± 1.1
**k_cat_** **(min^−1^)**	11.3 ± 1.3	16.0 ± 1.9	30.1 ± 4.2	43.2 ± 3.9
**k_cat_ / K_m_^Casein^** **(min^−1^ μM^−1^)**	2.5	4.1	6.3	10.8

^a^ All values represent the mean of three independent measurements ± SD.

**Table 3 ijms-21-06449-t003:** Kinetic parameters of CK1ε towards DDX3X and ATP.

Enzyme	Substrate	V_max_ (pmol min^−1^ μg^−1^)	K_m_ [μM]
CK1ε	DDX3X	0.8 ± 0.1 ^a^	0.2 ± 0.1 ^a^
CK1ε	ATP	1.7 ± 0.1	35.4 ± 8.5

^a^ All values represent the mean of three independent measurements ± SD.

**Table 4 ijms-21-06449-t004:** Kinetic parameter values of CK1ε as a function of varying ATP concentrations at fixed DDX3X amounts.

DDX3X [μM]	0.12	0.24	0.48	0.96
**V_max_^ATP^** **(pmol min^−1^** **μ** **g^−1^)**	0.9 ± 0.1 ^a^	1.3 ± 0.2 ^a^	1.9 ± 0.4 ^a^	2.1 ± 0.2 ^a^
**K_m_^ATP^** **[μM]**	26.7 ± 3.3	31.6 ± 12.5	34.9 ± 7.8	16.2 ± 6.0
**k_cat_** **(min^−1^)**	2.5 ± 0.3	3.7 ± 0.7	5.5 ± 1.0	6.0 ± 0.7
**k_cat_ / K_m_^ATP^** **(min^−1^ μM^−1^)**	0.09	0.11	0.16	0.37

^a^ All values represent the mean of three independent measurements ± SD.

**Table 5 ijms-21-06449-t005:** Kinetic parameter values of CK1ε as a function of varying DDX3X concentrations at fixed ATP amounts.

ATP [μM]	15	30	60	120
**V_max_^DDX3X^** **(pmol min^−1^** **μ** **g^−1^)**	1.2 ± 0.4 ^a^	1.5 ± 0.4 ^a^	2.9 ± 0.7 ^a^	4.6 ± 0.8 ^a^
**K_m_^DDX3X^** **[μM]**	0.6 ± 0.3	0.5 ± 0.2	0.7 ± 0.3	0.7 ± 0.2
**k_cat_** **(min^−1^)**	3.3 ± 1.2	4.2 ± 1.0	8.3 ± 2.0	13.2 ± 2.3
**k_cat_ / K_m_^DDX3X^** **(min^−1^ μM^−1^)**	5.5	8.4	11.9	18.9

^a^ All values represent the mean of three independent measurements ± SD.

**Table 6 ijms-21-06449-t006:** Representation of kinetic parameter values of CK1ε obtained as a function of varying ATP concentrations at fixed DVL2 amounts.

Dvl2 [μM]	0.03	0.06	0.12	0.18
**V_max_^ATP^** **(pmol min^−1^ μg^−1^)**	4.7 ± 0.8 ^a^	5.9 ± 1.0 ^a^	11.5 ± 1.5^a^	17.7 ± 1.8^a^
**K_m_^ATP^** **[μM]**	13.8 ± 3.8	7.8 ± 3.5	9.5 ± 3.2	6.7 ± 1.9
**k_cat_** **(min^−1^)**	13.6 ± 2.3	17.0 ± 2.9	33.1 ± 4.4	51.0 ± 5.1
**k_cat_ / K_m_^ATP^** **(min^−1^ μM^−1^)**	1.0	2.2	3.5	7.6

^a^ All values represent the mean of three independent measurements ± SD.

**Table 7 ijms-21-06449-t007:** Representation of kinetic parameter values of CK1ε obtained as a function of varying DVL2 concentrations at fixed ATP amounts.

ATP [μM]	7.5	30	60	120
**V_max_^Dvl2^** **(pmol min^−1^ μg^−1^)**	13.3 ± 2.8^a^	36.7 ± 23.1^a^	49.8 ± 27.3^a^	36.3 ± 4.4^a^
**K_m_^Dvl2^** **[μM]**	0.5 ± 0.1	1.1 ± 0.6	0.9 ± 0.5	0.4 ± 0.1
**k_cat_** **(min^−1^)**	38.2 ± 8.0	105.7 ± 64.3	143.4 ± 68.0	104.7 ± 12.6
**k_cat_ / K_m_^Dvl2^** **(min^−1^ μM^−1^)**	76.4	98.8	159.3	261.7

^a^ All values represent the mean of three independent measurements ± SD.

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
