# Peer review of "Novel Insights into the Biochemical Mechanism of CK1ε and its Functional Interplay with DDX3X"

_ijms, 2020, doi:10.3390/ijms21176449_

Round 1
Reviewer 1 Report
In this paper by Bono et al., the authors investigate the kinetic parameters of human CK1e on its substrates casein, DDX3X, and Dvl2. From these experiments, it is reported that DDX3X does not increase the phosphorylation of other CK1 substrates, as has been reported previously, but rather acts as a competing substrate itself. Further, phosphorylation by CK1 on the C-terminus of DDX3X increased the ssDNA-dependent ATPase activity of DDX3X.
There are major concerns with the evidence presented and its interpretation; these concerns are itemized below. The paper also does not include appropriate context or consideration of other published literature and the authors do not provide adequate explanations for why their data contradicts multiple other studies. Given these concerns, it is unlikely that this manuscript will alter the way the field considers the role of DDX3X as an activator of CK1e.
Major concerns
- How do the Km, Vmax, kcat, and kcat/Km values calculated here compare to other known kinetic constants for CK1? See, for example, S. pombe CK1 in Hoekstra et al. 1994, CK1d in Shinohara et al. 2017, and especially CK1e on B-TrCP peptide, CK1tide, and casein in Isojima et al. 2009. Along the same lines, how do the ID50 values for PF4800567 compare to what is already published?
- It is unclear why the data in Fig 2 are not consistent with that in Fig 1C and D. I would expect the ATP curve for casein at 3.1uM to resemble Fig 1D, and the casein curve for ATP at 30uM to resemble Fig 1C, but they do not, and the Km and Vmax values do not match either.
- According to Fig 3, DDX3X seems to be a poor substrate, as acknowledged in the discussion. But then how can DDX3X compete as a substrate with the more highly phosphorylated casein in Fig 7? Further discussion of these results is warranted.
- The first half of the paper is concerned with a kinetic analysis of CK1e. However, the hypotheses being tested and the mechanism that the data support are not clearly described. For example, the purpose of Figs 2, 4, and 6 is to determine whether the mechanism of phosphoryl transfer occurs via a ternary complex (that may be either ordered or random) or a ping-pong reaction, but this is never actually stated. Most kinases utilize a ternary complex, and when each substrate binds independently, i.e. when the concentration of ATP does not affect the affinity for casein, DDX3X, or Dvl2, this is consistent with random formation of the ternary complex. This is never explicitly stated; furthermore, contradictory explanations of these data occur in the results and the discussion – lines 219-220 say that ATP binds a CK1e-Dvl2 complex, but line 362 says casein binds a CK1e-ATP complex. Yet the data for all three protein substrates follow the same pattern. If there was a preferential order of substrate binding, and if that order differed with different protein substrates, what would the data be expected to look like? Including explicit descriptions of the hypothesis being tested and a model of the kinetic mechanism would aid in interpretation of these results.
- The kinetic information gained in this study is not used to advance the understanding of CK1 or DDX3X biology. There are no models or predictions formed about the cellular functioning of these proteins. Work on many other protein kinases shows that a random sequential mechanism is fairly common and expected, so what progress has been made biochemically? In vivo work that demonstrates the significance of this mechanism would address this concern.
- This work is not placed in the context of the published literature, which is especially important given that the findings contradict published work, which provide consistent results and interpretations across multiple laboratories. Data from Cruciat et al. 2013 and Dolde et al. 2018 support the model that physical interaction of DDX3X with CK1e promotes the phosphorylation of Dvl2, and this is also consistent with Guan et al. 2019, in which the LINC-mediated association of DDX3X and CK1e enhances Dvl2 phosphorylation. If the findings of this paper are correct, and in fact DDX3X binding does not enhance CK1e-mediated phosphorylation of Dvl in Wnt signaling, how can the role of the CK1e/DDX3X interaction in promoting Wnt signaling be explained? This needs to be addressed, ideally through an in vivo experiment, but at least in the discussion. Furthermore, Fig 7 shows competition between casein and DDX3X, but the Dvl2 experiments are not shown (lines 292-294). These experiments are vital, especially because they represent the physiologically relevant case, and they can be directly compared to the previous work.
- The effect of CK1e phosphorylation on DDX3X activity in Fig 8 also conflicts with previously published data. To test whether these differences are due to using ssDNA as a substrate, the experiment should be done with different substrates (dsRNA, ssRNA, ssDNA) side by side. The explanation given in the discussion about ds vs. ss nucleic acids is unclear given that Dolde et al. 2018 also used ssRNA, and rather it suggests that DNA does not behave in the same way as the physiological RNA substrate.
- There are several statements in the text that are not correct:
- Line 62 – CK1 is hardly a new molecular target, inhibitors have been in development for decades.
- Line 79 – Casein is not a natural substrate of CK1e.
- Line 137 – Saturation by both substrates is never achieved or tested, so how do you know this is required for maximal efficiency? ATP saturation is reached, but sufficiently high concentrations of protein substrate are not used, such that the curves at the two highest concentrations of ATP never level out for any of the three protein substrates.
- Line 259 – There is no evidence for interaction in this work.
- Line 329 – The inhibitor does not compete with DDX3X or casein binding, it competes with ATP binding.
Minor points
- The first paragraphs of the introduction and the discussion each contain several different ideas. It would be easier for the reader to understand if these were split up into smaller, more focused paragraphs.
- What was the concentration of enzyme used in Fig 1B?
- Lines 97-100 – These sentences are repeated.
- The data in Tables 1, 3, and 5 are already present in the text and are apparent from the graphs in Fig 1, 3, and 5. I would recommend placing the Km and Vmax values as insets on the graphs and deleting the tables.
- Statistics are needed for Fig 7 B & D. It would also be easier to read if the +/- DDX3X conditions were next to each other at each corresponding casein concentration.
- Line 274 – What are the amino acid residues for the truncated DDX3X?
- It is confusing that in Fig 8C there is no ATPase activity of CK1 alone, since the kinase autophosphorylates, as shown in Fig 7 and in previously published work. How much CK1 was used? This is not in the text, legend, or methods.
- Line 389 – Regarding autophosphorylation of CK1, you cannot state that there was no reduction in CK1 kinase activity without testing the activity of autophosphorylated vs. dephosphorylated kinase. I would expect that there is the same amount of autophosphorylation during the 15min period of the assay (see Fig 7), but this does not necessarily represent maximal activity of CK1.
- Line 434 – The “full length recombinant human active CK1e” should be described in more detail, and any caveats clearly stated. For example, this protein has an N-terminal GST tag, which would cause the kinase to dimerize. How could this affect the kinetic constants? The autophosphorylation state should also be addressed; most papers that use “active” CK1 truncate the C-terminus or dephosphorylate the enzyme prior to the kinase assay.

Author Response
Ms. ijms-877161 – R1
Answers to Reviewers
Reviewer 1:
Major concerns:
- How do the Km, Vmax, kcat, and kcat/Km values calculated here compare to other known kinetic constants for CK1? See, for example, S. pombe CK1 in Hoekstra et al. 1994, CK1d in Shinohara et al. 2017, and especially CK1e on B-TrCP peptide, CK1tide, and casein in Isojima et al. 2009. Along the same lines, how do the ID50 values for PF4800567 compare to what is already published?
We thank the reviewer for this point. Cross-comparison revealed significant differences in the reported values already among the three different studies cited by the Reviewer. This is not surprising, since the enzymes, substrates and reaction conditions were different. In the papers cited by the Reviewer not all the kinetic parameters as determined in our work were reported. Also, the reaction conditions were often quite different for our ones. However, there was a good agreement for example in the Km values for ATP, which were 24 µM in Hoekstra et al. 1994 for the yeast CK1 and 20 µM in Isojima et al. 2009 for human CK1e/d, as compared to the values reported in Table 2A. Also the kcat/Km values for ATP derived from the paper by Hoekstra et al. were similar to those reported by us (Table 2 A). Only Hoekstra 1994 and Shinohara 2017 reported the kinetic parameters for the protein substrate, which was, however, different: Hoekstra et al. used casein, while Shinohara et al. used a synthetic peptide. The kcat/Km values for casein as determined in our work (Table 2 B) were comparable to those reported by Hoekstra et al. for the yeast isoform Hhp1. The corresponding kcat/Km values reported by Shinohara et al. for CK1d on the synthetic peptide were 10 to 50-fold lower than those reported by us or by Hoekstra et al. In general, we can conclude that the kinetic parameters determined in our work for ATP were in agreement with those reported for other CK1 enzymes from yeast and human. The kinetic parameters for the protein substrates showed greater variability on the literature, but this can be ascribed to the different nature of the substrates utilized (protein vs. peptide), to the different enzymes (yeast and/or mammalian and different isoforms) and to different reaction conditions in vitro. However, even within this variability, the range of values was in the low-medium micromolar range for all the enzymes/studies analyzed, including ours.
The ID50 value for the inhibitor PF4800567 depends on the amount of the competing substrate ATP in the reaction. Walton et al., 2009 reported an ID50 of 32 nM against a synthetic peptide substrate, in the presence of 10 µM ATP. We reported an ID50 of 162 nM against casein, but in the presence of 30 µM ATP. By correcting for the competitive substrate concentration, the inhibitory potency (Ki) of PF4800567 in our tests was 71 nM, a value consistent with the one reported in literature, also considering the difference in the protein substrate utilized.
- It is unclear why the data in Fig 2 are not consistent with that in Fig 1C and D. I would expect the ATP curve for casein at 3.1uM to resemble Fig 1D, and the casein curve for ATP at 30uM to resemble Fig 1C, but they do not, and the Km and Vmax values do not match either.
We thank the Reviewer for this observation. The experiments in Fig. 1 and Table 1 were carried out in order to perform a preliminary setup of the reaction conditions and to determine the optimal range of concentrations of the various reagents (enzyme, substrates). Since the affinity and catalytic efficiency of the enzyme for each substrate (ATP, casein) is influenced by the concentration of the other substrate (casein, ATP) in the reaction, a certain variability might be expected between the experimental setting of Fig. 1 (single concentration of one substrate, varying the other) and the one of Fig. 2 (different fixed concentrations of one substrate, varying the other). Indeed the same discrepancies can be seen in the similar experiments of Fig. Fig. 5/Table 5. An additional source of variability was the fact that, since the experiments required the use of a significant amount of Ck1e enzyme, different batches of enzymes were used. Even though they all came from the same source and manufacturer, some variability could be present.
Even though the discrepancies noted above were of modest magnitude, we agree with the Reviewer that they could result in a less clear message of the manuscript. In addition, since the most precise determination of the kinetic parameters was the one performed with the variation of both substrates (Tables 2, 6) which all have been performed with the same batch of CK1e enzyme, we elected to remove Tables 1, and 5 and the corresponding panels from the Figures 1 and 5 from the revised manuscript and we modified the text accordingly.
- The first half of the paper is concerned with a kinetic analysis of CK1e. However,
the hypotheses being tested and the mechanism that the data support are not
clearly described. For example, the purpose of Figs 2, 4, and 6 is to determine
whether the mechanism of phosphoryl transfer occurs via a ternary complex (that
may be either ordered or random) or a ping-pong reaction, but this is never
actually stated. Most kinases utilize a ternary complex, and when each substrate
binds independently, i.e. when the concentration of ATP does not affect the
affinity for casein, DDX3X, or Dvl2, this is consistent with random formation of the
ternary complex. This is never explicitly stated; furthermore, contradictory explanations of these data occur in the results and the discussion – lines 219-220 say that ATP binds a CK1e-Dvl2 complex, but line 362 says casein binds a CK1e-ATP complex. Yet the data for all three protein substrates follow the same pattern. If there was a preferential order of substrate binding, and if that order differed with different protein substrates, what would the data be expected to look like? Including explicit descriptions of the hypothesis being tested and a model of the kinetic mechanism would aid in interpretation of these results.
We apologize for the lack of clarity of the text. We think our data point to a random mechanism of substrate binding. Indeed in the Discussion section (lines 357-359), we explicitly state: “The kinetic analysis performed in the first part of this project, indicated that CK1ε does not appear to follow an obligatory sequential binding of substrates.” In order to make it clearer, we have no changed it in the revised text into: “The kinetic analysis performed in the first part of this project, indicated that CK1ε appears to follow a random sequential binding of substrates.” We also agree with the unclear textual issues noted by the Reviewer and we have now harmonized all the text in order to avoid contradictory statements.
As for the kinetic model/hypothesis, the formation of a catalytically active ternary complex between an enzyme (E) and two substrates (A and B) could theoretically follow different kinetic pathways with different combinations of rate constants. For example, binding of substrate A can give a binary complex EA with equal, higher or lower affinity for substrate B with respect to the free enzyme. One could then expect to see equal, lower or higher Km values for the different substrates depending on the binary complex considered. This can be due to either a faster dissociation or a slower association of the substrates to the enzyme. We believe that it would make the manuscript too complex and overlong, to insert a scheme of all the possible kinetic pathways, since the data are consistent with a random model. We have however included some sentences in the text to better explain the kinetic model part.
- The kinetic information gained in this study is not used to advance the
understanding of CK1 or DDX3X biology. There are no models or predictions
formed about the cellular functioning of these proteins. Work on many other
protein kinases shows that a random sequential mechanism is fairly common and
expected, so what progress has been made biochemically? In vivo work that
demonstrates the significance of this mechanism would address this concern.
We agree with the Reviewer that in vivo experiments are needed to address the physiological functions of CK1e/DDX3X interaction. However, this was not the aim of our manuscript, whose goal was to provide an enzymatic characterization of DDX3X as a substrate for CK1e. We would like to note here that so far, to the best of our knowledge, no detailed kinetic analysis of DDX3X as a substrate of the kinase CK1e has been published in parallel with casein and Dvl2, with derivation of all the kinetic constants. We think that this represents and advancement in knowledge, with all the merits and limits of an in vitro approach.
- This work is not placed in the context of the published literature, which is
especially important given that the findings contradict published work, which
provide consistent results and interpretations across multiple laboratories. Data
from Cruciat et al. 2013 and Dolde et al. 2018 support the model that physical
interaction of DDX3X with CK1e promotes the phosphorylation of Dvl2, and this
is also consistent with Guan et al. 2019, in which the LINC-mediated association
of DDX3X and CK1e enhances Dvl2 phosphorylation. If the findings of this paper
are correct, and in fact DDX3X binding does not enhance CK1e-mediated
phosphorylation of Dvl in Wnt signaling, how can the role of the CK1e/DDX3X
interaction in promoting Wnt signaling be explained? This needs to be
addressed, ideally through an in vivo experiment, but at least in the discussion.
Furthermore, Fig 7 shows competition between casein and DDX3X, but the Dvl2
experiments are not shown (lines 292-294). These experiments are vital,
especially because they represent the physiologically relevant case, and they
can be directly compared to the previous work.
We thank the Reviewer for these important observations. We would like to note here that our data only suggest that DDX3X alone is not sufficient, in our full in vitro system, to stimulate phosphorylation of casein or Dvl2. This by no means implies that in vivo such stimulation would not take place. On the contrary, there are robust data suggesting that DDX3X acts both as a substrate and an activator for CK1e in Wnt signaling. Our hypothesis is that additional factors/co-factors acting in the cell are mediating such an activating role and we are planning to perform further investigations to clarify this point. In order to comply with the Reviewer’s request, we have now added the Dvl2 experiments in Suppl. Fig. 1. We have also better discussed our results in light of the published literature.
- The effect of CK1e phosphorylation on DDX3X activity in Fig 8 also conflicts with
previously published data. To test whether these differences are due to using
ssDNA as a substrate, the experiment should be done with different substrates
(dsRNA, ssRNA, ssDNA) side by side. The explanation given in the discussion
about ds vs. ss nucleic acids is unclear given that Dolde et al. 2018 also used
ssRNA, and rather it suggests that DNA does not behave in the same way as the
physiological RNA substrate.
We have previously published that DDX3X ATPase activity can also be stimulated by ssDNA (Garbelli et al.). In Dolde et al. 2018, Fig. 4 A and B, the ATPase assays were performed only with dsRNA. We agree with the Reviewer that a more detailed analysis with different substrates could have added more data, unfortunately we were not able to carry out these experiments due to the closure of the laboratories as a consequence of the national lockdown related to the Covid 19 emergency. We are nonetheless planning to expand the present work both in vitro and in vivo, including those experiments as suggested. But this will be the subject of a future manuscript.
- There are several statements in the text that are not correct:
- Line 62 – CK1 is hardly a new molecular target, inhibitors have been indevelopment for decades.
- Line 79 – Casein is not a natural substrate of CK1e
- Line 329 – The inhibitor does not compete with DDX3X or casein binding, itcompetes with ATP binding
We apologize for the imprecisions noted above. We have amended the revised text as requested by the Reviewer.
- Line 137 – Saturation by both substrates is never achieved or tested, so how
do you know this is required for maximal efficiency? ATP saturation is
reached, but sufficiently high concentrations of protein substrate are not used,
such that the curves at the two highest concentrations of ATP never level out
for any of the three protein substrates.
Ideally, substrate dependence curves for the determination of kinetic constants should have higher point densities in the region around the flex point. All the curves shown have a sufficient number of points both before and after the flex, to allow determination of the asymptote (Vmax) by the computer aided interpolation. The accuracy of the fitting is also shown by the good standard deviation of the calculated values. The only curve which is below saturation is the one with varying concentrations of Dvl2. The reason is that we were limited by the concentration of the commercially available stock solution, which did not allow us to use concentrations in the assay higher than those reported in the study. However, the fitting was still accurate enough to allow determination of the Vmax.
- Line 259 – There is no evidence for interaction in this work.
The Reviewer is correct in pointing out that we did not directly show a physical interaction between DDX3X and CK1e. However, we show that both DDX3X and Dvl2 are phosphorylated by CK1e, which necessarily implies physical interaction.
Minor points
The first paragraphs of the introduction and the discussion each contain several
different ideas. It would be easier for the reader to understand if these were split
up into smaller, more focused paragraphs
We thank the Reviewer for this suggestion. In the revised manuscript we tried to convey the same concepts in a more concise and focused manner.
- What was the concentration of enzyme used in Fig 1B?
All reactions with fixed CK1e concentration were performed in the presence of 2.5 ng/µl of recombinant enzyme. We have specified this in the revised text.
- Lines 97-100 – These sentences are repeated.
We have amended as requested.
- The data in Tables 1, 3, and 5 are already present in the text and are apparent
from the graphs in Fig 1, 3, and 5. I would recommend placing the Km and Vmax
values as insets on the graphs and deleting the tables.
We thank the Reviewer for this suggestion. As mentioned above, we have removed part of the Figures 1 and 5 and the corresponding Tables. However, for better clarity we would prefer to keep the other Tables as they are.
- Statistics are needed for Fig 7 B & D. It would also be easier to read if the +/- DDX3X conditions were next to each other at each corresponding casein concentration.
We thank the Reviewer for this suggestion. Statistics have been added in Fig. 7B, D and Supp. Fig. 1F. However, we would prefer to keep the order of data in the bar graphs as it is now, since we believe it can easily convey the message.
- Line 274 – What are the amino acid residues for the truncated DDX3X?
The N-DDX3X lacks the C-ter aa 428-662. We have now indicated it in the revised text as Δ(428-662).
- It is confusing that in Fig 8C there is no ATPase activity of CK1 alone, since the kinase autophosphorylates, as shown in Fig 7 and in previously published work. How much CK1 was used? This is not in the text, legend, or methods.
All reactions with fixed CK1e concentration were performed in the presence of 2.5 ng/µl of recombinant enzyme. The autoradiographic detection of incorporated 33P is more sensitive than the ATP-Glo assay used for ATPase. Form the autoradiographic images of Fig. 7 it is apparent that the incorporation of 33P due to autophosphorylation is low, so it is likely that the level of ATP hydrolysis due to autophosphorylation was below the detection limit of the ATPase assay.
- Line 389 – Regarding autophosphorylation of CK1, you cannot state that there was no reduction in CK1 kinase activity without testing the activity of autophosphorylated vs. dephosphorylated kinase. I would expect that there is the same amount of autophosphorylation during the 15min period of the assay (see Fig 7), but this does not necessarily represent maximal activity of CK1.
We thank the Reviewer for this remark. We have modified the sentence stating that under the conditions used, the autophosphorylation of CK1e was still compatible with an active enzyme.
- Line 434 – The “full length recombinant human active CK1e” should be described in more detail, and any caveats clearly stated. For example, this protein has an N-terminal GST tag, which would cause the kinase to dimerize. How could this affect the kinetic constants? The autophosphorylation state should also be addressed; most papers that use “active” CK1 truncate the C-terminus or
dephosphorylate the enzyme prior to the kinase assay.
We have used commercially available active full length recombinant human CK1ε expressed by baculovirus in Sf9 insect cells using an N-terminal GST tag (SignalChem, cat. no.C66-10G). All the reactions have been performed according to the manufacturer’s protocol. Since we did not express the protein ourselves, it was not possible to generate truncated version. Dephosphorylation of the enzyme prior to the kinase assay would have introduced an additional source of variability, since we could not guarantee always to achieve the same level of dephosphorylation, nor could we assume that the inactivation of the enzyme due to the incubation in the dephosphorylation reaction would have been always the same. On the other hand, with the untreated commercial enzyme, since our assays were performed under the same conditions, the basal activity of the enzyme could be compared among the assays, assuming the same level of autophosphorylation. The eventual dimerization due to the GST-tag should not affect significantly the kinetic reaction as long as the two monomers behave as independent enzymes or, theoretically, as an enzyme with two identical active sites. If the dimerization was generating an enzymatic form with non-equivalent substrate binding properties of the two monomers, positive or negative cooperativity should be expected. However, we did not observe any significant deviation in the kinetic curves from a standard Michaelis-Menten behavior.
Reviewer 2 Report
Introduction
The introduction is very clear and sufficient to understand the importance of the study the authors performed. In this work, authors better characterized CK1 epsilon activity in vitro:
- underlying a non-specific kinase activity of CK1 epsilon on DDX3X protein
- identifying a CK1epsilon stimulation of DDX3X activity
CK1 epsilon activity was recently reported to be affected in different diseases, such as cancer and neurodegeneration. Therefore, the importance of better characterizes its activity biochemically might be relevant to identify therapeutic drugs.
Minor revision
Results
The characterization of CK1 epsilon activity is very clear and detailed, I only suggest few Minor changes in the text and results:
Figure 2A and 2B on the text:
Line 117-…Vmax and Km values for the transfer reaction maintaining casein at fixed concentrations, while varying ATP (Figure 2A), and vice versa (Figure 2B).
I understand that each curve on the graph represents the situation described, but in the text is not so clear since the authors used different Casein concentration in 2A and ATP in 2B.
I think authors can explain it better to permit their understanding to a broader scientific readers’ and apply these modifications for each result where they calculate Vmax and Km.
Figure 7
I suggest adding statistics value for parts B and D, especially in figure D where authors conclude that in the presence or absence of N-DDX3X, casein phosphorylation does not change.
7C: could authors had also CK1 epsilon western blotting? Just to visualize its levels
Line 292- Similar experiments were performed… Could authors add these results, maybe as supplementary?
Discussion
Line 410. There is a ”DDX3X.” probably coming from an old sentence
Author Response
Ms. ijms-877161 – R1
Answers to Reviewers
Reviewer 2:
Minor points
Results
The characterization of CK1 epsilon activity is very clear and detailed, I only suggest few Minor changes in the text and results:
Figure 2A and 2B on the text: Line 117-…Vmax and Km values for the transfer reaction maintaining casein at fixed concentrations, while varying ATP (Figure 2A), and vice versa (Figure 2B). I understand that each curve on the graph represents the situation described, but in the text is not so clear since the authors used different Casein concentration in 2A and ATP in 2B. I think authors can explain it better to permit their understanding to a broader scientific readers’ and apply these modifications for each result where they calculate Vmax and Km.
We thank the Reviewer for this suggestion. The confusion comes from the fact that in all cases both substrates are varied, but in a different way. When we say “fixed concentrations” of substrate A, we mean that we perform titration of substrate B in the presence of a fixed dose of A, but we repeat this for a range of concentrations of A. In other words, it is the same as carrying out several substrate B titrations, each in the presence of a different concentration of A. We understand that the concept might be a bit convolute and in the revised version we specified better what was the design of the experiment.
Figure 7: I suggest adding statistics value for parts B and D, especially in figure D where authors conclude that in the presence or absence of N-DDX3X, casein phosphorylation does not change.
We thank the Reviewer for this suggestion, statistics have been added as requested
7C: could authors had also CK1 epsilon western blotting? Just to visualize its levels
We thank the Reviewer for this suggestion, western blots have been added as requested
Line 292- Similar experiments were performed… Could authors add these results, maybe as supplementary?
We thank the Reviewer for this suggestion, the Dvl2 experiment has been added in supp. Fig. 1 as requested
Discussion
Line 410. There is a ”DDX3X.” probably coming from an old sentence
We apologize for the oversight, the text has been corrected as requested